# Nutritional Issues in Children with Congenital Heart Diseases (CHDs)

**DOI:** 10.3390/nu17243936

**Published:** 2025-12-16

**Authors:** Giovanna Fernanda Vazzana, Alessia Romano, Claudio Romano

**Affiliations:** 1Pediatric Gastroenterology and Cystic Fibrosis Unit, Department of Human Pathology in Adulthood and Childhood “G. Barresi”, University Hospital “G. Martino”, 98124 Messina, Italy; giadavazzana@gmail.com; 2Pediatric Unit, University Tor Vergata, 00133 Rome, Italy; alessiarom99@gmail.com

**Keywords:** congenital heart diseases, malnutrition, paediatric nutrition, feeding difficulties, micronutrients, perioperative care

## Abstract

Infants and children with congenital heart diseases (CHDs) are especially susceptible to malnutrition. The pathogenesis of nutritional disorders in this population reflects a multifactorial balance between increased metabolic demands, reduced dietary intake, and altered nutrient absorption. This narrative review summarizes current knowledge on the prevalence, risk factors, and underlying mechanisms of malnutrition in paediatric CHDs. It also discusses available tools for nutritional assessment, practical strategies for dietary management, and perioperative considerations. Early identification through screening, combined with individualized nutritional support and multidisciplinary care, is essential to optimize growth and enhance clinical recovery. Special attention is given to feeding difficulties, micronutrient imbalances, and the contribution of human milk and energy-dense feeding approaches in promoting growth. Integrating nutrition into cardiac management is essential to improve both short- and long-term outcomes. Future research should focus on the development of standardized, evidence-based protocols and the implementation of precision nutrition in paediatric CHDs.

## 1. Introduction

Malnutrition can negatively impact growth, surgical outcomes, and neurodevelopment in children with congenital heart diseases (CHDs) [1]. CHDs are among the most frequent congenital anomalies, affecting approximately 0.8–1.2% of live births worldwide [2]. Advances in diagnosis and management have markedly improved survival: over 85% of affected children now reach adulthood, and under-five mortality has declined significantly [2]. Despite these developments, approximately one-third of children still have hemodynamically significant congenital heart defects that need to be treated [3].

Malnutrition is highly prevalent in children with CHDs, with reported global rates ranging from 15% to 64% [4]. A recent meta-analysis found pooled preoperative prevalences of underweight 27.4%, stunting 24.4%, and wasting 24.8% [5]. Cohort studies further indicate that over one-third of children may be underweight, wasted, or stunted, with key risk factors including low birth weight, prematurity, pulmonary hypertension, and pneumonia [6]. The highest risk is observed in children under one year of age and in those with anaemia, pulmonary hypertension, or congestive heart failure, the latter being associated with a sixfold increase in malnutrition risk [7].

The aetiology of malnutrition in CHDs is multifactorial. Metabolic demands are increased by cardiac factors, including increased energy expenditure, volume and/or pressure overload, myocardial dysfunction, chronic hypoxemia, congestive heart failure, pulmonary hypertension, and pulmonary overflow [8]. Extracardiac contributors, including chromosomal abnormalities, gastroesophageal reflux, intestinal malabsorption, prematurity, recurrent infections, and intrauterine growth restriction, further exacerbate nutritional deficits [9,10]. Feeding difficulties, malabsorption, and certain medications, such as diuretics, may additionally impair nutrient intake and assimilation [9,10].

Although most infants with CHDs are born with normal weight, growth and nutritional deficits often manifest within the first months of life, particularly in moderate-to-severe cases [5]. The type and severity of CHDs strongly influence malnutrition risk: lower-risk lesions include early operated patent ductus arteriosus, atrial septal defect, and pulmonary stenosis, whereas high-risk defects include pulmonary atresia, tetralogy of Fallot, complex septal defects, transposition of the great arteries, hypoplastic left heart syndrome, and other severe anomalies (see Table 1 for CHDs’ classification and related malnutrition risk in children) [11].

Malnutrition may contribute to short-term and long-term adverse outcomes, including longer hospital length of stay, higher risk of infection, higher mortality rates, higher risk of adverse neurodevelopmental outcomes, and poorer quality of life [5].

Therefore, early identification, close monitoring, and timely nutritional interventions are crucial to improving outcomes. Nevertheless, current practices remain heterogeneous, highlighting the need for standardized, evidence-based guidelines [12].

Given the high prevalence and multifactorial nature of malnutrition in this population, this narrative review focuses on children with CHDs. It aims to provide a comprehensive overview of nutritional issues in children with CHDs, including their prevalence, risk factors, underlying mechanisms, and management strategies. Additionally, we provide a nutritional management flow chart designed for use beyond the intensive care setting. By incorporating guidelines, highlighting useful anthropometric and screening tools, and offering an organized framework for nutritional management, this review seeks to close current gaps. It provides a thorough resource to support tailored nutritional care in children with CHDs by fusing evidence-based recommendations with practical clinical guidance, enhancing and expanding earlier consensus documents and guideline-focused reviews.

By synthesizing current evidence, we aim to identify knowledge gaps and inform future research and clinical practice.

## 2. Methods

A literature search was conducted in PubMed, Scopus, and Cochrane using keywords including “malnutrition”, “congenital heart diseases”, “nutritional management”, and “paediatric nutrition”, including articles published up to 2025. Studies were selected based on their relevance to nutritional status, growth, risk factors, and management strategies in children with CHDs. Both observational and interventional studies published in English were considered. Given the narrative nature of this review, literature selection was guided by relevance rather than a formal systematic process, and the heterogeneity of CHDs phenotypes and predominance of observational data are acknowledged as limitations.

## 3. Pathophysiology of Nutritional Disorders in Paediatric CHDs

Malnutrition arises from a complex interplay of increased metabolic demands, reduced nutrient intake, and impaired absorption mechanisms strongly linked to the underlying cardiac pathology and heart failure [13]. Understanding these processes is critical to improving growth, immune function, and overall cardiac outcomes in this vulnerable population.

### 3.1. Increased Metabolic Demands

Children with CHDs and heart failure typically exhibit elevated energy requirements due to chronic hypoxemia, increased work of breathing, and heightened cardiac workload [4]. Cardiac insufficiency and hypoxia substantially raise resting energy expenditure, while recurrent infections, surgical stress, and systemic inflammation further exacerbate catabolic stress [4,11]. These children often require significantly more energy to maintain homeostasis and support growth.

### 3.2. Reduced Nutritional Intake

Feeding difficulties are common and multifactorial. Tachypnea, fatigue, and poor oral-motor coordination limit effective feeding, particularly in infants. Postoperative factors such as sedation, mechanical ventilation, and gastrointestinal dysmotility further impair intake. Psychosocial issues, including caregiver anxiety, early breastfeeding interruption, or delayed complementary feeding, may compound these challenges [11].

### 3.3. Malabsorption and Inefficient Nutrient Utilization

Even with adequate intake, nutrient assimilation and utilization may be impaired. Low cardiac output compromises intestinal perfusion, leading to mucosal oedema and reduced absorption. Cyanotic CHDs and Fontan circulation can cause mesenteric ischemia and protein-losing enteropathy, resulting in loss of albumin, immunoglobulins, and fat-soluble vitamins [14]. Hepatic congestion further distorts nutrient metabolism and storage, compounding nutritional deficits.

### 3.4. Clinical Implications

In children with CHDs, nutritional deficits result from a complex interplay of increased metabolic requirements, reduced dietary intake, and malabsorption. Micronutrient deficiencies—particularly iron, vitamin D, and carnitine—are common and may exacerbate growth failure [14,15]. Severe protein-energy malnutrition can impair cardiac structure and function, contributing to myocardial atrophy, reduced ventricular mass, and increased risk of arrhythmias [16]. Early and repeated hospitalizations, low birth weight, and pulmonary hypertension further amplify the risk of undernutrition [7].

Collectively, these findings emphasize the importance of systematic nutritional screening and timely dietary intervention in paediatric cardiac care [15].

## 4. Nutritional Screening and Assessment Tools in Children with CHDs

### 4.1. Timing and Screening Tools

Systematic nutritional screening is now recognized as a standard of care in paediatric cardiology. All children with CHDs should undergo nutritional evaluation within 24 h of diagnosis or hospital admission, with reassessment at each clinical transition, including pre and postoperative stages [15]. Early identification of nutritional risk is essential, as malnutrition can significantly impact morbidity, recovery, and long-term outcomes [17].

For children with prolonged hospitalizations exceeding two weeks, weekly reassessments are strongly recommended to promptly detect evolving nutritional risks [15]. Children classified as at moderate or high risk of malnutrition should receive a comprehensive nutritional assessment conducted by trained healthcare professionals [17,18].

Validated screening tools for initial evaluation include Paediatric Yorkhill Malnutrition Score (PYMS), Screening Tool for Risk on Nutritional status and Growth in Kids (STRONGkids), Screening Tool for the Assessment of Malnutrition in Paediatrics (STAMP), and Paediatric Nutrition Screening Tool (PNSS) [19,20]. PYMS demonstrates high sensitivity and predictive value for identifying malnutrition risk in paediatric patients with CHDs, showing strong agreement with dietitian assessments and anthropometric standards [21]. It performs optimally in inpatient settings where nurse-led screening is feasible, although its moderate specificity may overestimate nutritional risk in children, particularly in the presence of oedema or cardiomyopathy-related alterations in body composition [22]. PYMS is validated for use in children aged 1 to 16 years [21]. STRONGkids is robust in predicting poor clinical outcomes such as long hospital stay, increased readmissions and emergency department visits in hospitalized children, including those with congenital heart disease [22]. Its simplicity supports implementation across diverse clinical settings, although its lower specificity reduces accuracy in complex diseases [22]. There is no consensus for a single gold standard, but combining PYMS for risk identification and STRONGkids for outcome prediction is supported by recent validation studies [23]. Subjective Global Nutritional Assessment (SGNA) offers higher specificity and more comprehensive clinical information, making it well-suited for specialized outpatient cardiology programs [24]. Despite its lower sensitivity and reliance on subjective components, SGNA is preferred for detailed assessment due to its reliability and strong correlation with clinical outcomes [24]. In paediatric cardiology, it is typically applied after initial risk screening to provide a thorough evaluation of malnutrition severity and to guide individualized nutritional interventions. STRONGkids and SGNA are both validated for use in infants and children up to 18 years [23,24].

### 4.2. Anthropometric Assessment

Recommended anthropometric monitoring strategies for paediatric patients with CHDs include serial measurement of weight, length (recumbent for infants), and head circumference for infants, with calculation and interpretation of Z-scores for weight-for-age, length/height-for-age, and weight-for-length/height [25,26]. For infants younger than six months and preterm infants, growth should be plotted on the World Health Organization (WHO) growth charts for term infants and the Fenton 2013 growth charts for preterm infants until term-equivalent age, after which transition to WHO charts is appropriate [27]. Z-scores provide a standardized assessment of growth status and severity of malnutrition, with a decrease of ≥1 in Z-score indicating growth faltering [26].

Weight-for-age Z-score (WAZ) reflects overall nutritional status, length/height-for-age Z-score (LAZ/HAZ) identifies stunting, and weight-for-length Z-score (WLZ/WHZ) detects wasting or overweight [27]. These indices are critical for risk stratification and perioperative planning, as lower WAZ and HAZ are associated with increased morbidity and mortality after cardiac surgery [26]. Each 1-unit decrease in WAZ z-score is associated with 33% higher adjusted odds of operative mortality (aOR 1.33, 95% CI 1.25–1.41) in children undergoing congenital heart surgery [26]. Head circumference should be monitored in infants, as it correlates with neurodevelopmental outcomes and may parallel declines in other growth indices in children with CHDs [28].

Supplementary measures, such as mid-upper arm circumference (MUAC) and skinfold thickness, are valuable for assessing muscle and fat stores, especially when standard measures are confounded by fluid shifts or oedema [29]. MUAC is validated for children 6–59 months and correlates with BMI and nutritional risk [30]. The timing of surgical or interventional procedures, feeding methods, and cardiac diagnosis are important factors to take into account. Infants with complex or cyanotic heart disease, pulmonary hypertension, or single-ventricle physiology are at the highest risk for growth failure [29]. Serial assessments every 3–6 months, complemented by body composition measures by a multidisciplinary team, are recommended, with individualized interventions based on growth trends, feeding tolerance, and surgical timing, particularly for higher-risk patients, in accordance with current guidelines and expert consensus [24,25]. Importantly, post-intervention follow-up is essential, as children may exhibit “catch-up growth” following corrective or palliative procedures [15].

In summary, optimal growth assessment in paediatric CHDs requires serial anthropometry, Z-score interpretation using WHO and Fenton charts, supplementary measures for body composition, and individualized nutritional strategies tailored to cardiac diagnosis, feeding modality, and procedural timing, while monitoring for catch-up growth after interventions (see Table 2) [25].

### 4.3. Clinical Evaluation

While anthropometry provides objective growth data, it does not capture the full complexity of nutritional challenges in children with CHDs. A comprehensive clinical assessment should include evaluation for signs of heart failure, such as tachypnoea, hepatomegaly, peripheral oedema, and feeding fatigue [31]. Assessment of feeding performance—including suck strength, duration of feeding, meal frequency, tolerance, vomiting, diarrhea, and recurrent infections—provides critical insight into caloric adequacy and gastrointestinal function [32]. In children with cyanotic CHDs, monitoring oxygen saturation fluctuations during feeding, cyanotic episodes, and fatigue can indicate nutritional compromise [33,34]. Additionally, a detailed dietary intake history, including calories consumed, volumes tolerated, and feeding modalities, is essential for individualized nutritional planning [35].

### 4.4. Laboratory Investigations and Micronutrient Assessment

Laboratory evaluations play an essential role in the comprehensive assessment of children with CHDs, complementing clinical examination and anthropometric measurements [36]. These evaluations provide critical insights into biochemical markers of malnutrition, which may not be fully captured by growth charts alone. Routine investigations commonly include complete blood count, renal and hepatic function tests, iron studies, electrolytes, albumin, prealbumin, thyroid function, and micronutrients such as zinc, copper, and magnesium [37]. Interpretation of protein markers, including albumin and prealbumin, requires careful consideration, as changes may reflect underlying cardiac or hepatic dysfunction in addition to nutritional deficits [38].

Micronutrient deficiencies are frequent in children with CHDs, particularly affecting calcium, iron, folate, and vitamin B12, which underscores the need for individualized supplementation strategies [38]. Recent literature further highlights the complexity of these deficiencies. For instance, inadequate dietary intake of iron, zinc, vitamin E, calcium, and magnesium has been documented in children with CHDs, correlating closely with malnutrition risk and growth impairment [37]. Preoperative underweight, stunting, and wasting are also highly prevalent and may exacerbate laboratory abnormalities and micronutrient deficits [38]. Folate deficiency is particularly concerning, with evidence of deficiency present in both affected infants and their mothers, emphasizing the importance of pre- and postnatal nutritional support [39]. In low-resource settings, anaemia is highly prevalent among children with cyanotic CHDs, highlighting the critical need for routine monitoring of haemoglobin and iron status [40,41].

Considering these findings, laboratory assessment should ideally include an extended panel of micronutrients—iron, folate, vitamin B12, zinc, copper, magnesium, vitamin D, and potentially selenium—especially for children presenting with clinical or anthropometric signs of malnutrition or those receiving chronic therapies that may impair nutrient absorption or increase losses [38]. Clinicians must interpret laboratory results in the context of hemodynamic burden, hepatic congestion, chronic hypoxia, and systemic inflammation, all of which may independently affect biomarkers. Consequently, serial monitoring and trend analysis are recommended over single time-point measurements to guide targeted nutritional interventions and optimize outcomes [15].

## 5. Nutritional Requirements and Intervention Strategies

### 5.1. Energy and Protein Requirements

Children with complex or critical CHDs often exhibit increased energy expenditure compared with healthy peers, which can complicate growth and nutritional recovery [42]. Feeding difficulties, swallowing impairments, and complications such as gastro-oesophageal reflux are common, and their management requires a multidisciplinary approach involving cardiologists, dietitians, and speech and feeding specialists [15].

Whenever possible, resting energy expenditure (REE) should be measured by indirect calorimetry to accurately guide energy provision; in its absence, predictive equations such as Schofield or WHO are acceptable alternatives [18]. Nutritional requirements must be individualized according to disease severity, nutritional risk, lesion type, hemodynamic burden, feeding tolerance, fluid restrictions, and comorbidities.

Expert consensuses suggest stratified caloric and protein intakes: children at low nutritional risk may require approximately 90–100 kcal/kg/day with 1.5 g protein/kg/day, whereas moderate-risk children may benefit from 110–120 kcal/kg/day with 2.5 g protein/kg/day [10,11,43]. In high-risk infants with hemodynamically significant CHDs and malnutrition, energy requirements may reach 120–150 kcal/kg/day with up to 4 g protein/kg/day, and in some cases, caloric needs may approach three times the basal metabolic rate [10,19,44]. These values underscore the importance of tailored nutritional prescriptions to support adequate growth and metabolic balance (see Table 3).

Successful nutritional management requires a dynamic, multidisciplinary approach. In children who cannot tolerate adequate oral intake due to fatigue, malabsorption, or fluid restrictions, supplemental enteral feeding via nasogastric or gastrostomy tubes may be necessary, and parenteral nutrition should be considered when enteral feeding is insufficient [45].

Children with acquired heart disease, including cardiomyopathy, myocarditis, or post-operative heart failure, may face similar nutritional challenges. Increased energy expenditure, protein loss due to chronic illness, and pharmacological therapies such as diuretics or inotropes further increase protein and caloric requirements [46].

Nutritional interventions must also adapt to the phase of the disease. Preoperative nutrition focuses on early identification and correction of malnutrition, optimization of energy and protein intake, nutritional prehabilitation, and minimized fasting, with enteral or parenteral support provided when oral intake is insufficient [14]. During acute illness or post-operative critical phases, energy intake should initially match measured or predicted REE to avoid metabolic overload, with gradual increases during recovery to promote catch-up growth [19]. Post-discharge, energy provision of approximately 140–150 kcal/kg/day for children aged 0–36 months may be appropriate to achieve optimal growth trajectories, in line with expert consensus [43,47]. Catch-up growth after intervention in children with CHDs varies by lesion complexity. Children with simple CHDs generally achieve rapid and near-complete catch-up growth, while those with complex CHDs, particularly single ventricle physiology, show slower, less complete, and more variable recovery, with persistent deficits in weight, length, and head circumference [19].

In summary, individualized, dynamic, and closely monitored nutritional strategies are essential for all children with CHDs to ensure adequate growth, support metabolic demands, and improve clinical outcomes.

### 5.2. Feeding Modalities and Strategies

Breastfeeding remains the preferred feeding modality in infants with CHDs, given its immunological advantages, improved feeding adaptation, and association with favourable outcomes in complex cardiac conditions [48]. When human milk is unavailable or insufficient, standard infant formulas or follow-up formulas are recommended, whereas semi-elemental or hydrolysed formulas enriched with medium-chain triglycerides may be indicated in cases of malabsorption or gastrointestinal compromise [49]. Recent consensus statements emphasise the prioritisation of human milk even in infants with severe CHDs due to its protective effect against infection and superior feeding tolerance [48,49].

In the context of fluid restriction—common in heart-failure states or significant hemodynamic burden—energy needs can be met through high-density feeds. Strategies include adding modular components, cautiously introducing complementary foods (generally not before four months of age), or using hypercaloric polymeric formulas ≥1 kcal/mL, based on expert consensus [11,50]. These approaches should be implemented with close monitoring, as increased caloric density may elevate insensible water losses by 10–15% or more, particularly in the presence of tachypnoea or oedema [11,51]. Human milk–based and energy-dense feeding strategies show promising benefits for growth and postoperative outcomes in children with CHDs, but evidence remains limited by study heterogeneity and scarce multicentre trials [49]. Marked variability in clinical practice highlights the need for rigorous prospective multicentre studies to standardize feeding protocols and confirm their clinical effectiveness.

Oral feeding remains preferable in hemodynamically stable infants to support physiological swallowing, oral-motor development, and parent–infant interaction. When full oral intake is insufficient—owing to feeding fatigue, respiratory compromise, or inadequate weight gain—a combined regimen of oral feeding supplemented with nocturnal nasogastric tube (NGT) feeding may be used; establishing NGT feeding during hospitalisation ensures adequate training and monitoring of safety and tolerance [50].

Percutaneous endoscopic gastrostomy (PEG) is indicated in children with persistent feeding difficulties or growth failure when oral or NGT feeding is inadequate [52]. PEG provides reliable long-term enteral access, reduces complications associated with prolonged NGT use, and improves growth outcomes, including in high-risk populations such as hypoplastic left heart syndrome [53]. PEG placement is generally safe in CHDs when performed by experienced teams, with feeding typically initiated within hours in stable patients [52,53]. Evidence, derived largely from retrospective cohorts, demonstrates that PEG is feasible across a broad range of cardiac phenotypes, including single-ventricle physiology [54]. Although associated with improved postoperative growth velocity, moderate-to-severe malnutrition may persist at one year, indicating that early nutritional gains may not fully compensate for chronic preoperative deficits and warranting continued nutritional surveillance [53].

PEG-related complications are generally minor, including granulation tissue, local erythema, and tube malfunction; major complications are rare [52]. In neonates with CHDs, gastrostomy placement is associated with higher readmission and device-related complication rates compared with non-cardiac infants, although laparoscopic techniques appear to mitigate these risks [55]. Long-term dependence on PEG is common, especially in children with significant preoperative feeding impairment, prolonged hospitalisation, or neurodevelopmental comorbidities [56]. Early PEG placement has been associated with lower BMI in later childhood, suggesting vulnerability of long-term growth trajectories despite adequate early caloric delivery [57].

Parenteral nutrition is reserved for infants in whom enteral feeding is impossible or insufficient—such as those with severe malabsorption, prolonged chylothorax, or critical hemodynamic instability—and must follow established paediatric critical-care protocols [15]. In CHDs, parenteral nutrition requires careful attention to fluid balance, energy provision, and metabolic stability to avoid complications while supporting recovery and growth.

In summary, optimal feeding strategies in infants and children with CHDs require an individualised, adaptive approach integrating the benefits of human milk, the strategic use of high-density feeds in fluid-restricted states, combined oral–enteral regimens when needed, and the judicious use of PEG (see Table 4) or parenteral nutrition. Such tailored interventions are central to supporting growth, maintaining metabolic stability, and improving clinical outcomes.

### 5.3. Fluid and Electrolyte Management

In children with CHDs, fluid and electrolyte management is crucial due to hemodynamic compromise, diuretic therapy, and altered water and sodium balance. Total daily fluid intake is generally recommended not to exceed approximately 165 mL/kg/day, with sodium intake limited to 2.2–3 mEq/kg/day to prevent fluid overload and worsening heart failure [11]. Achieving the optimal balance is essential: excessive fluid can exacerbate oedema and heart failure, whereas overly restrictive intake may compromise tissue perfusion, growth, and renal function, particularly in neonates, infants, and postoperative patients [51]. According to observational studies, dysnatremias are common in infants following heart surgery, with hyponatremia linked to positive fluid balance and hypotonic fluids and hypernatremia frequently associated with transfusions and low free water intake. [58]. This highlights the need for individualized fluid therapy, preferential use of isotonic solutions, and careful monitoring of fluid balance, electrolytes, and daily weight. In post-operative patients or those with increased ADH secretion, fluid intake must be carefully titrated to avoid overload while maintaining adequate perfusion (see Table 5) [11,51].

### 5.4. Micronutrient Deficiencies and Supplementation

Micronutrient deficiencies are frequent in children with CHDs due to chronic inflammation, reduced intake, blood losses, or prolonged hospitalization. Iron deficiency is particularly common and contributes to anaemia, worsening hypoxemia, and increased cardiac workload [59]. Iron supplementation should be guided by ferritin, transferrin saturation, and haemoglobin levels, with standard paediatric dosing of 3–6 mg/kg/day of elemental iron for treatment and 1–2 mg/kg/day for prophylaxis in high-risk patients, continuing for 3–6 months after haemoglobin normalization [15,60]. In cyanotic CHDs, ferritin may be falsely elevated, making transferrin saturation and reticulocyte haemoglobin content more reliable indicators [61].

Vitamin D deficiency is common and may contribute to impaired bone mineralization, delayed wound healing, increased infection risk, and altered myocardial contractility [62]. Supplementation is recommended to maintain serum 25-hydroxyvitamin D levels ≥ 80 nmol/L, especially in high-risk children [62,63]. Preoperative vitamin D deficiency has been associated with higher vasoactive-inotropic scores and longer mechanical ventilation in children with CHDs [62].

Other micronutrients, including magnesium, zinc, selenium, carnitine, and fat-soluble vitamins A, E, and K, may also be depleted, particularly in children receiving prolonged enteral or parenteral nutrition or those with protein-losing enteropathy [38]. Carnitine is essential for myocardial energy metabolism, and deficiency—secondary to chronic illness or certain medications—can impair cardiac function; supplementation is recommended in confirmed cases [64]. Magnesium, zinc, and phosphate replacement should also be considered postoperatively or in children on long-term nutritional support, with adjustments guided by laboratory monitoring [38].

Overall, careful and individualized management of fluids, electrolytes, and micronutrients is essential to support growth, optimize cardiac function, and improve outcomes in children with CHDs.

## 6. Special Considerations

### 6.1. Cyanotic Versus Non-Cyanotic CHDs

Children with CHDs, such as those with single-ventricle physiology, typically experience more severe growth impairment, higher metabolic demands, and greater feeding difficulties than children with non-cyanotic defects [65]. These patients often exhibit increased resting energy expenditure due to chronic hypoxemia, elevated cardiac workload, and altered muscle metabolism [66]. Tailored nutritional plans are therefore essential to meet individual metabolic requirements and prevent further deterioration of nutritional status.

In cyanotic CHDs, chronic hypoxia triggers adaptive erythropoiesis, leading to increased red blood cell mass and elevated haematocrit levels. However, this compensatory mechanism can coexist with iron deficiency, resulting in paradoxical polycythaemia characterized by increased blood viscosity but reduced oxygen-carrying efficiency [67]. Careful assessment of ferritin, transferrin saturation, and reticulocyte haemoglobin is required to guide iron therapy. Iron supplementation should be individualized and adjusted to avoid excessive erythrocytosis, which may exacerbate hyperviscosity and compromise tissue perfusion [67].

### 6.2. Single-Ventricle and Fontan Physiology

Children with single-ventricle physiology and those who have undergone Fontan palliation represent a particularly high-risk group for nutritional compromise. Post-Fontan patients frequently develop protein-losing enteropathy (PLE) and lymphatic dysfunction, which can lead to hypoalbuminemia, fat malabsorption, and micronutrient deficiencies [68]. These complications often require specialized nutritional strategies aimed at optimizing energy intake while reducing lymphatic flow.

In single-ventricle populations, exclusive human milk feeding is associated with substantially improved outcomes compared with lower or no human milk exposure, including a 72% reduction in the odds of postoperative necrotizing enterocolitis (OR 0.28, 95% CI 0.15–0.50), a 71% reduction in sepsis (OR 0.29, 95% CI 0.13–0.65), and a 25% shorter length of hospital stay (rate ratio 0.75, 95% CI 0.66–0.86) [48].

Formulas enriched with MCT are recommended in cases of chylothorax or fat malabsorption, as they bypass intestinal lymphatic transport and provide an efficient energy source [69]. In children with significant PLE, MCT-based feeds help reduce lymphatic congestion and improve protein retention, supporting better growth and metabolic recovery. Omega-3 fatty acid supplementation may be considered according to general paediatric recommendations for cardiovascular health, though routine use of probiotics is not currently supported by strong evidence [69,70].

Long-term success in the management of children with complex cyanotic or single-ventricle physiology relies on continuous caregiver education, multidisciplinary follow-up, and individualized nutritional plans. Coordination among cardiologists, dietitians, and gastroenterologists ensures timely recognition of nutritional deterioration and early implementation of targeted interventions, which are critical to improving both quality of life and long-term outcomes.

## 7. Monitoring and Follow-Up

Children with CHDs frequently require enteral nutrition via NGT or PEG, particularly after complex cardiac surgery. Over half of infants with complex CHDs need tube feeding at initial postoperative discharge, with some remaining dependent for several months [71]. Risk factors for prolonged feeding include single ventricle physiology, vocal cord impairment, perioperative aspiration risk, and comorbid genetic syndromes [71,72]. PEG is indicated for persistent feeding difficulties or growth failure not manageable with oral or NGT feeding [73]. Removal is considered after sustained oral intake and adequate growth, usually following 8–12 weeks of non-use and multidisciplinary review [52,74]. Weaning from tube feeding should be individualized and guided by multidisciplinary assessment. Criteria include consistent oral intake of 50–75% of caloric needs over at least 48 h, evidence of stable growth, and absence of aspiration risk [75]. The process involves evaluation of oral-motor skills, swallowing safety, gradual advancement of oral feeds, and continued monitoring for feeding intolerance or cardiorespiratory compromise. Supplemental tube feeds are maintained until oral intake is adequate. Long-term PEG in children with CHDs can affect caregiver well-being in complex ways [74]. As feeding stabilizes, many families report an improvement in their quality of life, despite the initial anxiety and burden. [73]. Social isolation, irregular routines, sleep disturbances, and stigma are examples of persistent problems. Structured education, multidisciplinary support, and peer mentoring are key to addressing caregiver needs and promoting both nutritional outcomes and psychosocial well-being.

Dysphagia and oral/visceral hypersensitivity are common, particularly after cardiac surgery, and are more prevalent in children with single ventricle physiology or vocal cord injury [76]. Management requires structured evaluation, including instrumental swallow studies, oral-motor therapy, and graduated progression of textures. Feeding therapy targets oral-motor coordination, sensory modulation, and safe swallowing. Interventions for persistent intolerance may include adjustments in caloric density, feed volume, and anti-reflux strategies, with ongoing parental education and psychosocial support [72]. Structured nutritional follow-up is essential to ensure the efficacy and safety of interventions. During hospitalization, daily monitoring of weight, fluid balance, and feeding tolerance allows early detection of changes that may compromise hemodynamic stability [33]. After discharge, growth assessments should occur every 2–4 weeks until catch-up growth is achieved, then every 3–6 months, with serial anthropometric evaluation using disease-specific growth curves [11]. Laboratory monitoring—including complete blood count, ferritin, electrolytes, and vitamin D every 4–6 weeks—guides supplementation, particularly in patients on chronic diuretics or with cyanotic lesions [15,33]. Persistent weight loss, feeding intolerance, or abnormal laboratory findings should prompt multidisciplinary reassessment [11]. High-risk patients benefit from weekly reviews focusing on growth velocity, feeding tolerance, and metabolic balance, with follow-up intervals extended progressively as clinical stability improves [11,15].

Nutritional status is a key determinant of long-term neurodevelopment and quality of life in children with CHDs [77]. Early malnutrition can contribute to cognitive, language, and executive function deficits, along with poorer academic and psychosocial outcomes. Individualized nutritional support, addressing growth failure and micronutrient deficiencies, can foster catch-up growth and improve neurodevelopment [78]. Biomarker-guided precision nutrition is a promising approach to optimize growth and clinical outcomes in children with CHDs. Key actionable biomarkers can include high-sensitivity C-reactive protein, interleukin-6, tumor necrosis factor-alpha, insulin, adiponectin, leptin, uric acid, N-terminal pro-B-type natriuretic peptide, and selected lipidomic and metabolomic signatures, which reflect systemic inflammation, metabolic dysregulation, and cardiac stress [79]. None of these biomarkers can be considered specific for patients with CHDs. Preliminary studies suggest that nutritional interventions tailored to these biomarkers can improve metabolic resilience and patient outcomes, although prospective multicentre trials are needed to establish standardized protocols and confirm clinical efficacy.

In summary, effective monitoring and follow-up in children with CHDs demand a customized strategy that combines structured feeding evaluation and therapy with anthropometric and laboratory evaluations. This approach minimizes problems associated with artificial nutrition and feeding challenges while optimizing growth, nutritional status, and perioperative outcomes.

## 8. Conclusions

Malnutrition is a frequent and clinically relevant comorbidity in children with CHDs, contributing to increased morbidity, prolonged hospitalization, and poorer outcomes. Cardiac dysfunction, altered metabolism, and feeding difficulties interact to create a state of nutritional vulnerability that requires early recognition and tailored intervention [11,15].

Assessment should extend beyond growth chart interpretation to include clinical and functional parameters. Signs such as fatigue, feeding intolerance, vomiting, diarrhea, and recurrent infections often indicate nutritional compromise and warrant prompt evaluation [11]. Laboratory assessment provides additional insight, identifying deficiencies in micronutrients such as iron, zinc, copper, magnesium, and vitamin D [15,33].

The pathophysiology of malnutrition in CHDs is multifactorial. Increased metabolic demands, reduced oral intake from fatigue or tachypnea, and impaired nutrient absorption contribute to growth failure [15]. A proactive approach—combining early screening, individualized dietary planning, and regular monitoring—is essential for optimal outcomes (see Figure 1). Children showing faltering growth or declining Z-scores should be referred early to a dietitian for detailed assessment and intervention [19,21].

Micronutrient deficiencies remain widespread and clinically significant. Regular monitoring and correction of trace elements and vitamins are necessary to prevent adverse effects on myocardial function, immune response, and wound healing [11].

Ultimately, addressing nutritional issues in paediatric CHDs requires an integrated, multidisciplinary approach. Cardiologists, dietitians, and paediatricians should collaborate to ensure continuity of nutritional care and implement evidence-based protocols aimed at preventing malnutrition and improving recovery trajectories. Within this framework, the concept of precision nutrition is increasingly relevant, emphasizing risk stratification that integrates cardiac and nutritional parameters, individualized nutritional targets based on clinical phenotype or metabolic features, and, where available, the use of disease-specific growth curves. These components show the possibility of more specialized interventions that match nutritional management with the variety of paediatric CHDs, even though they are still in the early stages of routine practice.

## Figures and Tables

**Figure 1 nutrients-17-03936-f001:**
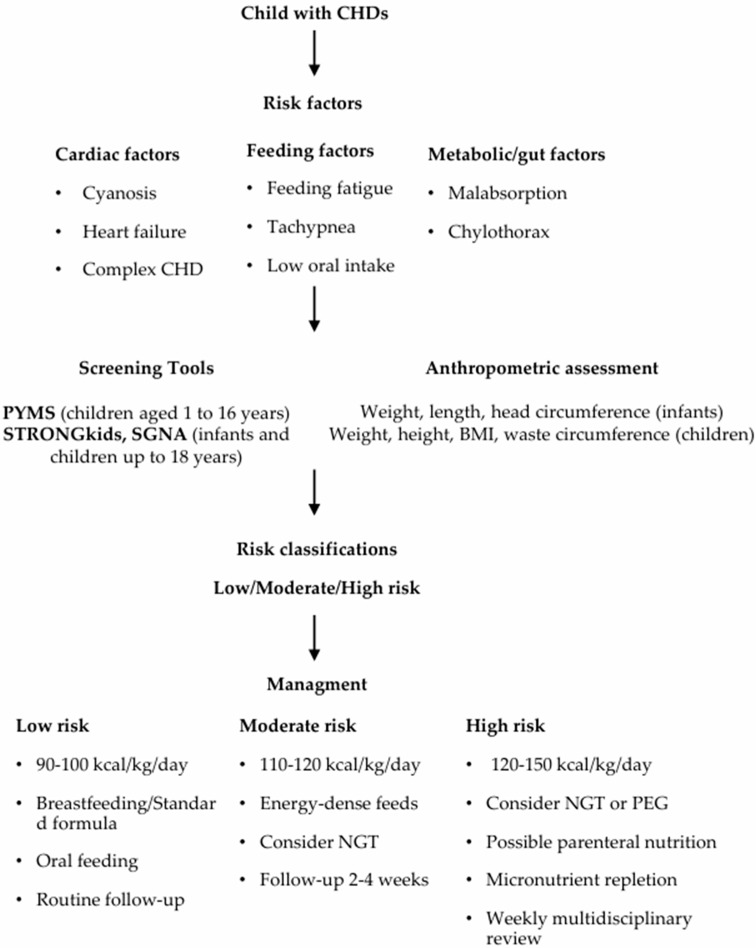
Proposed Flowchart for the Management of Children with CHDs.

**Table 1 nutrients-17-03936-t001:** CHDs’ classification and related malnutrition risk in children.

Classification	Features	Common Examples	Risk of Malnutrition in Children
**Cyanotic** **heart diseases**	Right-to-left shunt; systemic hypoxemia; clinical cyanosis	Tetralogy of Fallot; Transposition of the Great Arteries; Tricuspid Atresia; Total Anomalous Pulmonary Venous Return; Pulmonary Atresia; Hypoplastic Left Heart Syndrome	High risk, especially with pulmonary hypertension, heart failure, or delayed diagnosis
**Non-cyanotic** **heart diseases**	Left-to-right shunt or obstructive lesions; absence of cyanosis	Ventricular Septal Defect; Atrial Septal Defect; Patent Ductus Arteriosus; Coarctation of the Aorta; Pulmonary Stenosis; Aortic Stenosis	Moderate to high risk; increases with heart failure, pulmonary hypertension, or significant structural disease

**Table 2 nutrients-17-03936-t002:** Summary of anthropometric assessment in children with CHDs.

	Infants and Toddlers	Children
**Recommended anthropometric measurements**	Weight Length Head circumference	Weight, Height BMI Waist circumference
**Z-scores**	WAZ Z-score WLZ/WHZ Z-score LAZ/HAZ Z-score	WAZ Z-score WLZ/WHZ Z-score LAZ/HAZ Z-scores BMI Z-score
**Growth reference charts**	WHO standards for term infants; Fenton charts for preterm infants until term-equivalent age.	
**Supplementary anthropometry**	MUAC (6–59 months) Triceps/subscapular skinfolds	
**Cardiac and metabolic factors**	Cardiac physiology, heart failure, pulmonary hypertension, feeding difficulties, and increased energy expenditure influence growth interpretation.	
**Monitoring and follow-up**	Serial assessments every 3–6 months; consider feeding modality and post-surgical recovery for “catch-up” growth.	

**Table 3 nutrients-17-03936-t003:** Summary of energy and protein requirements in children with complex or critical CHDs.

Nutritional Risk	Energy Requirements	Protein Requirements	Assessment Method	Clinical Considerations
**Low risk**	90–100 kcal/kg/day	1.5 g/kg/day	Predictive equations (Schofield, WHO)	Mild disease; stable growth; minor hemodynamic burden
**Moderate risk**	110–120 kcal/kg/day	2.5 g/kg/day	Indirect calorimetry preferred; predictive equations acceptable	Moderate disease severity; some feeding difficulties; increased energy expenditure
**High risk/severe disease**	120–150 kcal/kg/day (up to 3× basal metabolic rate in some infants)	Up to 4 g/kg/day	Indirect calorimetry strongly recommended	Hemodynamically significant lesions; malnutrition; complex CHDs; requires intensive nutritional support

**Table 4 nutrients-17-03936-t004:** Feeding Modalities and Strategies in Children with CHDs.

Feeding Strategy	Indications	Essential Considerations
**Human Milk**	Stable or postoperative infants	Preferred option; optimal tolerance and immune advantages
**Standard/Follow-up Formula**	When human milk is insufficient or unavailable	Suitable for most infants; monitor growth response
**Semi-elemental/Hydrolysed Formula**	Malabsorption or gastrointestinal compromise	Enhanced digestibility; supports nutrient absorption
**High-Density Feeds**	Fluid restriction; elevated caloric needs	≥1 kcal/mL; monitor hydration and tolerance closely
**Oral + NGT Feeding**	Feeding fatigue, respiratory burden, inadequate weight gain	Oral intake supplemented by nocturnal NGT feeding
**NGT**	Temporary enteral support	Short-term solution; may cause irritation if prolonged
**PEG**	Persistent feeding difficulty or growth failure	Long-term stable access; supports growth; minor complications frequent
**PN**	Severe malabsorption or hemodynamic instability	Use when enteral feeding is not feasible; requires strict monitoring

**Table 5 nutrients-17-03936-t005:** Fluid and electrolyte management in children with CHDs.

Indicator	Key Recommendations
**Fluid intake**	Total daily fluid intake generally ≤165 mL/kg/day; careful titration required in neonates, infants, and post-operative patients to avoid overload or underperfusion.
**Sodium intake**	Limit to 2.2–3 mEq/kg/day to prevent fluid overload and worsening heart failure.
**Monitoring**	Daily weight, fluid balance, and electrolytes; adjust therapy based on hemodynamic status and renal function.
**Electrolyte disturbances**	Dysnatremias common post-cardiac surgery: hypernatremia linked to transfusions and low free water; hyponatremia associated with positive fluid balance and hypotonic fluids.
**Fluid type considerations**	Prefer isotonic solutions; individualized adjustments based on ADH secretion, cardiac output, and clinical status.

## Data Availability

The original contributions presented in this study are included in the article. Further inquiries can be directed to the corresponding author.

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
