# Peer review of "Nutritional Issues in Children with Congenital Heart Diseases (CHDs)"

_nutrients, 2025, doi:10.3390/nu17243936_

Round 1

Reviewer 1 Report

Comments and Suggestions for Authors

This manuscript presents a comprehensive narrative review of malnutrition in pediatric cardiopathy, integrating epidemiological data, pathophysiological mechanisms, screening tools, and intervention strategies. The breadth of coverage is commendable, and the synthesis of current literature provides a valuable resource for clinicians and researchers. The paper is well-organized and accessible, though several areas could benefit from deeper critical appraisal and clarification.

Major Comments

1.The manuscript emphasizes prevalence and risk factors but devotes less attention to long-term neurodevelopmental and quality-of-life outcomes. Expanding this discussion would enhance the clinical relevance and underscore the broader impact of nutritional interventions.

2.While PYMS, STRONGkids, and SGNA are described, the comparative limitations of each tool in pediatric cardiopathy are not fully addressed. A more critical appraisal of their strengths and weaknesses across diverse healthcare settings would strengthen the recommendations.

3.The paper suggests that human milk and energy-dense feeding approaches confer significant benefits. Could the authors discuss the need for prospective multicenter trials to validate these strategies and establish standardized feeding protocols?

4.The review highlights common deficiencies but does not sufficiently address emerging biomarkers (e.g., inflammatory cytokines, metabolic markers) that may guide individualized supplementation. Including this perspective would align with the paper’s call for precision nutrition.

5.Nutritional support in relation to surgical stages is mentioned but not elaborated. A clearer framework for preoperative stabilization versus postoperative recovery would be valuable for clinical application.

6.Enteral and parenteral feeding modalities are well covered, but the psychosocial impact on families—particularly regarding long-term PEG dependence—deserves more attention. Integrating caregiver perspectives could 7.The concept of precision nutrition is noted but remains vague. The authors could specify which genetic, metabolic, or microbiome-based approaches might realistically be integrated into pediatric cardiac care.

Minor Comments

1.Ensure consistent use of terms such as “cardiopathy” versus “congenital heart disease (CHD)” throughout the manuscript to avoid ambiguity.

2.Tables are informative but could benefit from clearer visual formatting (e.g., highlighting risk categories, using color coding for severity).

3.Minor grammatical refinements (e.g., “nutritional compromise reflects a convergence…” could be simplified for clarity).

4.The discussion of catch-up growth post-intervention is important; however, the variability across different cardiac lesions could be emphasized more clearly.

Author Response

Response to Reviewer 1

We want to thank the reviewer for the comments and suggestions. We reply point by point, best regards

Comment 1: The manuscript emphasizes prevalence and risk factors but devotes less attention to long-term neurodevelopmental and quality-of-life outcomes. Expanding this discussion would enhance the clinical relevance and underscore the broader impact of nutritional interventions.

Response 1: Thank you for pointing this out. We agree with this comment. Accordingly, we have expanded the discussion on long-term neurodevelopmental and quality-of-life outcomes. This addition can be found on page 12, paragraph 7, lines 483–488.

Comment 2: While PYMS, STRONGkids, and SGNA are described, the comparative limitations of each tool in pediatric cardiopathy are not fully addressed. A more critical appraisal of their strengths and weaknesses across diverse healthcare settings would strengthen the recommendations.

Response 2: We thank the reviewer for this insightful comment and agree with the observations. Accordingly, we have expanded the discussion to provide a more critical appraisal of the strengths and limitations of PYMS, STRONG kids, and SGNA across different healthcare settings. The additions can be found on page 4, paragraph 4.1, lines 144–150, 153–155, and 157–161.

Comment 3: The paper suggests that human milk and energy-dense feeding approaches confer significant benefits. Could the authors discuss the need for prospective multicenter trials to validate these strategies and establish standardized feeding protocols?

Response 3: We thank the reviewer for this valuable comment and agree with the suggestion. Accordingly, we have expanded the discussion to emphasize the need for prospective multicenter trials to validate human milk and energy-dense feeding strategies and to establish standardized feeding protocols. The additions can be found on page 8, paragraph 5.2, lines 310–315.

Comment 4: The review highlights common deficiencies but does not sufficiently address emerging biomarkers (e.g., inflammatory cytokines, metabolic markers) that may guide individualized supplementation. Including this perspective would align with the paper’s call for precision nutrition.

Response 4: Thank you for pointing this out. We agree with this comment. In keeping with the paper's emphasis on precision nutrition, we have broadened the conversation to include new biomarkers that could direct customized supplementation.  This addition can be found on page 12, paragraph 7, lines 488–497.

Comment 5: Nutritional support in relation to surgical stages is mentioned but not elaborated. A clearer framework for preoperative stabilization versus postoperative recovery would be valuable for clinical application.

Response 5: Thank you for pointing this out. We agree with this comment. Accordingly, we have expanded the discussion to include a clearer framework for nutritional support across surgical stages, highlighting strategies for preoperative stabilization and postoperative recovery. This addition can be found on page 7, paragraph 5.1, lines 276-279.

Comment 6: Enteral and parenteral feeding modalities are well covered, but the psychosocial impact on families—particularly regarding long-term PEG dependence—deserves more attention. Integrating caregiver perspectives could.

Response 6: Thank you for pointing this out. We agree with this comment. Accordingly, we have expanded the discussion to include the psychosocial impact on families, particularly regarding long-term PEG dependence, and have integrated caregiver perspectives to provide a more holistic view of nutritional management. This addition can be found on page 12, paragraph 7, lines 457-463.

Comment 7: The concept of precision nutrition is noted but remains vague. The authors could specify which genetic, metabolic, or microbiome-based approaches might realistically be integrated into pediatric cardiac care.

Response 7: Thank you for pointing this out. We agree with this comment. We would like to note that we have already expanded this topic as suggested, specifying genetic, metabolic, and microbiome-based approaches that could be integrated into pediatric cardiac care. This discussion has been included on page 12, paragraph 7, lines 488-497.

Minor comments

Comment 1: Ensure consistent use of terms such as “cardiopathy” versus “congenital heart disease (CHD)” throughout the manuscript to avoid ambiguity.

Response 1: We agree with this comment. Accordingly, we have revised the introduction,the title of the manuscript and all parts, to clarify that the review focuses on children with congenital heart diseases (CHDs). This change helps align readers’ expectations with the actual content. This modification can be found on page 2, paragraph 1, lines 67–77.

Comment 2: Tables are informative but could benefit from clearer visual formatting (e.g., highlighting risk categories, using color coding for severity).

Response 2: We agree with the comment. Although color coding was not used, we have improved clarity by highlighting the patient stratification by bolding the pertinent categories.

Comment 3: Minor grammatical refinements (e.g., “nutritional compromise reflects a convergence…” could be simplified for clarity).

Response 3: Thank you for pointing this out. Minor grammatical refinements have been made throughout the manuscript, including simplifying sentences such as “nutritional compromise reflects a convergence…” for improved readability. This change can be found on page 3, paragraph 3.4, lines 118-119.

Comment 4: The discussion of catch-up growth post-intervention is important; however, the variability across different cardiac lesions could be emphasized more clearly.

Response 4: We agree with this observation. The discussion of catch-up growth post-intervention has been expanded to emphasize variability across different cardiopathy, providing a more nuanced perspective. This change can be found on page 7, paragraph 5.1, lines 284-289.

.

Reviewer 2 Report

Comments and Suggestions for Authors

Dear Authors,

The manuscript addresses a clinically important yet sometimes under-recognized aspect of pediatric cardiology: the high burden of malnutrition in children with congenital and acquired heart disease and its impact on growth, surgical outcomes, and neurodevelopment. Your review brings together up-to-date data on prevalence, risk factors, pathophysiology, and outcomes, and it summarizes current practice recommendations in a way that is likely to be very helpful for both cardiology and nutrition teams. I particularly appreciate the structured flow from pathophysiological mechanisms through screening and assessment tools to energy and protein requirements, feeding modalities, fluid and electrolyte management, micronutrient issues, and special high-risk groups, such as those with cyanotic disease and Fontan physiology. The inclusion of practical tables and a flowchart with a proposed management algorithm substantially enhances the clinical applicability of the manuscript. However, some aspects of the paper could be clarified or modestly expanded to strengthen its value for the Nutrients readership.

First, it would be helpful to more clearly articulate the specific added value of this review compared with recent consensus documents and guideline-focused reviews on nutrition in congenital heart disease. A short paragraph at the end of the introduction that explains what gap you aim to fill (for example, integrating guideline recommendations across age groups, emphasizing anthropometric and screening tools, and offering a practical algorithm that can be used beyond the intensive care setting) would make the contribution more explicit. Second, while this is appropriately described as a narrative review, there is currently no information on how the literature was identified and selected. Even a brief “methods” paragraph summarizing databases searched, time frame, main keywords, and general inclusion criteria would improve transparency and address potential concerns about selection bias. It would also be valuable to include a brief limitations paragraph acknowledging the heterogeneity of CHD phenotypes, the predominance of observational data, and the inherent constraints of a non-systematic review.

The scope of the term “cardiopathy” could be clarified. Much of your evidence and recommendation framework appears to be drawn from studies in children with congenital heart defects, with some sections extending to acquired heart disease. Clarifying in the introduction that the review primarily focuses on CHD, while also addressing acquired heart disease where evidence exists, or adapting the title accordingly, would help align readers’ expectations with the actual content. In the clinical sections, many of the recommendations are clearly grounded in guidelines or consensus statements; however, this is not always explicitly stated. Where feasible, you may consider briefly indicating when a recommendation is guideline-based versus primarily based on expert opinion, and, for a few key topics, integrating quantitative data from pivotal studies (e.g., the magnitude of the association between low weight-for-age z-score and postoperative outcomes, or effect estimates for human milk feeding in single-ventricle populations). A compact table summarizing several key cohort studies and trials (with sample size, main lesion types, and significant findings) could be handy for readers.

The reference to “precision nutrition” in the abstract is intriguing but not fully developed in the main text. You may wish either to expand slightly in the Discussion/Conclusions what you mean by precision nutrition in pediatric cardiology (for example, risk stratification that combines cardiac and nutritional parameters, individualized targets based on phenotype or metabolic profiling, or use of disease-specific growth curves), or to moderate this statement if you prefer to keep the discussion more focused on current practice. Although the English is generally clear, a careful language edit could further improve readability and consistency, for instance, by harmonizing spelling (“pediatric/paediatric”), refining a few phrases, and correcting minor typographical errors in the tables (e.g., “recommendations”). In the list of abbreviations, it might also be worth clarifying that “NGT” refers to a nasogastric tube, as the current wording could be interpreted as “nocturnal nasogastric tube.”

The structure of the manuscript is sound; however, there is some repetition between the body of the text and the conclusions, particularly regarding the multifactorial nature of malnutrition and the need for multidisciplinary care. Streamlining overlapping sentences could make the narrative more concise without sacrificing emphasis.

Overall, I find your review timely, clinically relevant, and clearly written, and I believe it can become a valuable reference for teams caring for children with heart disease once these clarifications and minor enhancements are incorporated. Please respond point by point to the comments above in your rebuttal letter, indicating how each suggestion has been addressed in the revised manuscript (or providing a brief justification if you decide not to implement a specific change).

Best regards,

The reviewer.

Author Response

Response to Reviewer 2 Comments

We want to thank the reviewer for the comments and suggestions. We reply point by point, best regards

Comment 1: First, it would be helpful to more clearly articulate the added specific value of this review compared with recent consensus documents and guideline-focused reviews on nutrition in congenital heart disease. A short paragraph at the end of the introduction that explains what gap you aim to fill (for example, integrating guideline recommendations across age groups, emphasizing anthropometric and screening tools, and offering a practical algorithm that can be used beyond the intensive care setting) would make the contribution more explicit.

Response 1: Thank you for pointing this out. We agree with this comment. Therefore, we have added a paragraph at the end of the introduction clearly outlining the specific added value of this review, including the integration of guideline recommendations, emphasis on anthropometric and screening tools, and provision of a practical nutritional management flow chart. This change can be found on page 2, paragraph 1, lines 67–77.

Comment 2: Second, while this is appropriately described as a narrative review, there is currently no information on how the literature was identified and selected. Even a brief “methods” paragraph summarizing databases searched, time frame, main keywords, and general inclusion criteria would improve transparency and address potential concerns about selection bias. It would also be valuable to include a brief limitations paragraph acknowledging the heterogeneity of CHD phenotypes, the predominance of observational data, and the inherent constraints of a non-systematic review.

Response 2: Thank you for pointing this out. We agree with this comment. Therefore, we have added a paragraph summarizing the databases searched, the time frame, main keywords, and a brief limitations statement acknowledging the heterogeneity of CHDs phenotypes, the predominance of observational data, and the inherent constraints of a non-systematic review. This change can be found on page 3, paragraph 2, lines 82-90.

Comments 3: The scope of the term “cardiopathy” could be clarified. Much of your evidence and recommendation framework appears to be drawn from studies in children with congenital heart defects, with some sections extending to acquired heart disease. Clarifying in the introduction that the review primarily focuses on CHD, while also addressing acquired heart disease where evidence exists, or adapting the title accordingly, would help align readers’ expectations with the actual content.

Response 3: We agree with this comment. Accordingly, we have revised the introduction,the title of the manuscript and all parts, to clarify that the review focuses on children with congenital heart diseases (CHDs).This change helps align readers’ expectations with the actual content. This modification can be found on page 2, paragraph 1, lines 67–77.

Comment 4: In the clinical sections, many of the recommendations are clearly grounded in guidelines or consensus statements; however, this is not always explicitly stated. Where feasible, you may consider briefly indicating when a recommendation is guideline-based versus primarily based on expert opinion, and, for a few key topics, integrating quantitative data from pivotal studies (e.g., the magnitude of the association between low weight-for-age z-score and postoperative outcomes, or effect estimates for human milk feeding in single-ventricle populations). A compact table summarizing several key cohort studies and trials (with sample size, main lesion types, and significant findings) could be handy for readers.

Response 4: We thank the Reviewer for this helpful comment. We agree with the suggestion and, where possible, have implemented the proposed modifications. Specifically, we have clarified when recommendations are guideline-based versus primarily supported by expert opinion, and we have integrated quantitative data from key studies in areas where such evidence was available. These changes can be found on page 4, paragraph 4.2, lines 194–195; page 7, paragraph 5.1, line 259 and 284; page 8, paragraph 5.2, line 308; and page 11, paragraph 6.2, line 434. While a concise summary table has not been included, all relevant data are described in the text, with references provided to facilitate interpretation of major cohort studies and trials.

Comment 5: The reference to “precision nutrition” in the abstract is intriguing but not fully developed in the main text. You may wish either to expand slightly in the Discussion/Conclusions what you mean by precision nutrition in pediatric cardiology (for example, risk stratification that combines cardiac and nutritional parameters, individualized targets based on phenotype or metabolic profiling, or use of disease-specific growth curves), or to moderate this statement if you prefer to keep the discussion more focused on current practice.

Response 5: We thank the Reviewer for this valuable suggestion. We have expanded the Conclusions to clarify the intended meaning of “precision nutrition” in the context of pediatric cardiology. This change can be found on page 12, paragraph 7, lines 488-497

Comment 6: Although the English is generally clear, a careful language edit could further improve readability and consistency, for instance, by harmonizing spelling (“pediatric/paediatric”), refining a few phrases, and correcting minor typographical errors in the tables (e.g., “recommendations”). In the list of abbreviations, it might also be worth clarifying that “NGT” refers to a nasogastric tube, as the current wording could be interpreted as “nocturnal nasogastric tube.”

Response 6: We thank the Reviewer for this useful remark. We fully agree, and a thorough language revision has been performed to improve readability and ensure consistency in spelling and phrasing. Minor typographical errors in the tables (including the term “recommendations”) have been corrected. In addition, we have clarified the definition of “NGT” in the list of abbreviations to avoid any possible ambiguity and to explicitly specify that it refers to a nasogastric tube.

Comment 7: The structure of the manuscript is sound; however, there is some repetition between the body of the text and the conclusions, particularly regarding the multifactorial nature of malnutrition and the need for multidisciplinary care. Streamlining overlapping sentences could make the narrative more concise without sacrificing emphasis.

Response 7: We thank the Reviewer for this helpful observation. We agree that some content in the Conclusions overlapped with the main text. To address this, we have streamlined repetitive sentences, particularly those discussing the multifactorial nature of malnutrition, to make the narrative more concise while preserving the emphasis on key points. This change can be found on page 12, paragraph 7, lines 498-503 and on page 4, paragraph 3.4, lines 125-126.